# Intercomparison of Three Open-Source Numerical Flumes for the Surface Dynamics of Steep Focused Wave Groups



**Thomas Vyzikas** [1,2,*] **, Dimitris Stagonas** [3] **, Christophe Maisondieu** [2] **and Deborah Greaves** [1]

1   School of Engineering, Computing and Mathematics, University of Plymouth, Plymouth PL4 8AA, UK;
    deborah.greaves@plymouth.ac.uk
2   Ifremer Centre Bretagne, Laboratoire Comportement des Structures en Mer, 29280 Plouzané, France;
    Christophe.Maisondieu@ifremer.fr
3   Department of Civil and Environmental Engineering, University of Cyprus, Nicosia 1678, Cyrpus;
    stagonas@ucy.ac.cy
*   Correspondence: thomas.vyzikas@gmail.com

**Abstract:** NewWave-type focused wave groups are commonly used to simulate the design wave for a given sea state. These extreme wave events are challenging to reproduce numerically by the various Numerical Wave Tanks (NWTs), due to the high steepness of the wave group and the occurring wave-wave interactions. For such complex problems, the validation of NWTs against experimental results is vital for confirming the applicability of the models. Intercomparisons among different solvers are also important for selecting the most appropriate model in terms of balancing between accuracy and computational cost. The present study compares three open-source NWTs in OpenFOAM, SWASH and HOS-NWT, with experimental results for limiting breaking focused wave groups. The comparison is performed by analysing the propagation of steep wave groups and their extracted harmonics after employing an accurate focusing methodology. The scope is to investigate the capabilities of the solvers for simulating extreme NewWave-type groups, which can be used as the "design wave" for ocean and coastal engineering applications. The results demonstrate the very good performance of the numerical models and provide valuable insights to the design of the NWTs, while highlighting potential limitations in the reproduction of specific harmonics of the wave group.

**Keywords:** extreme waves; focused waves; dispersion; harmonic analysis; CFD; OpenFOAM; SWASH; HOS

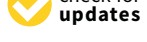



## 1. Introduction

The existence of abnormally large waves was mainly based on anecdotes from mariners, who referred to them as "walls of water", "holes in the sea", "three sisters" and "mad dogs" [1]. Hard evidence for the presence of unexpectedly large waves in the ocean, which are commonly referred to as extreme waves, came with measurements of the surface elevation at offshore platforms [2].

To present, a universal scientific consensus on the definition of extreme waves and their inclusion in the design process are yet to be reached. There are threshold criteria used to characterize extreme waves, but they are set to an extent empirically and say little about the underlying physics that distinguish extremes from normal large waves. This distinction was examined in a recent analysis of 122 million individual waves measured in the field, showing that there are several similarities between the average shape of extreme waves and that of normal very large waves [3]. Thus, from an engineering design perspective, the distinction between very large and extreme waves may be less relevant, and what is required to model these waves is the spectral characteristics of the design storm and its return period. The former can be found from historic data and the use of spectral models for hindcasting and forecasting. Then, a series of random-phase simulations can be performed for finding the largest wave events that the design storm can give at the location of interest.

A simpler way to do this was suggested by Tromans et al. [4], who used a probabilistic analysis to demonstrate that the average shape of very large waves for a given sea state approaches a "global form" that can be estimated analytically. This analysis is known as the NewWave theory and it computes the average shape of large waves as the scaled autocorrelation function of the underlying linear spectrum. The advantage of NewWave is its deterministic nature, which minimizes the need for random simulations. Consequently, NewWave is used extensively in experimental and numerical wave flumes for representing the largest waves in the ocean in the form of focused wave groups. NewWave also allows to select the most appropriate "design wave" for the examined marine structure by adjusting the relative phasing of the wave components in the group to simulate the greatest possible impacts on the structure [5].

Nonetheless, the simulation of NewWave-type groups also entails challenges, especially for cases with high wave steepness and nonlinearity. This nonlinearity triggers the emergence of both resonant and bound wave interactions in short propagation distances of a few wavelengths [6–9]. The bound waves create a locally steeper wave group than that predicted by linear theory, and the resonant interactions alter the dispersive properties of the components of the wave group, resulting in considerable changes in the amplitudes and phases of the underlying free waves. Similar effects are also observed for directional sea states where the energy spreads to higher frequencies and laterally to the main direction as a result of resonant interactions [10]. A consequence of these changes is the loss of symmetry of the shape of the wave group at focus, which results in a less steep wave, with apparent subsequent issues in determining the maximum impact on the examined structure. To improve the quality of the wave focusing, various empirical methodologies have been suggested that correct the phases of the wave group. Even after applying these methodologies, temporal and spatial shifts of the produced results are still commonly necessary to determine the location and time of the maximum crest of the focused wave [9,11]. To tackle these shortcomings, the focusing methodology of Stagonas et al. [12] is used in the present study, which can focus accurately the phases and amplitudes of the wave components. Part of the methodology is the decomposition of the wave signal into its harmonics, which further allows for more in depth observations of the physical behaviour of the dispersion of the wave group, which is a core element of the present study.

*What solvers are used in the design practice?* Despite the scientific progress in the understanding of large waves, the engineering design practice still lags behind, as highlighted in the recent position paper of DNV GL [13]. To change this, further studies for the propagation of steep waves are required, which will support the industry in revising the present practices with the aim to improve safety at sea. Traditionally, studies of steep waves were performed experimentally, but nowadays that powerful numerical solvers have emerged, NWTs have become a norm [14]. These solvers mainly employ the basic governing equations of fluid flows, namely the Navier-Stokes Equations (NSE), with appropriate boundary conditions for wave generation and absorption. The applicability of these NWTs depends on the assumptions and simplifications at the governing equations and the complexity of the numerical methods employed. The selection of the most suitable NWT relies on the complexity of the examined problem and the available resources. Computational Fluid Dynamics (CFD) solvers are considered the gold standard in wave modelling, being able to resolve complex fluid structure interactions problems and overturning waves. However, the main issue with CFD solvers is the high computational cost, which restricts them from operational coastal and ocean engineering applications. For such cases, faster solvers can be employed for the wave propagation, but they are limited to weakly nonlinear phenomena with parameterized or no wave breaking. These models can be used in combination with the computationally expensive CFD solvers for reducing the overall cost when large domains are considered, as a part of an integrated modelling system [15,16]. A critical point of passing down information from one model to the other refers to the accurate reproduction of the wave signal itself.

The scope of the present work is to investigate the performance of the operational model SWASH [17] and the pseudo-spectral model HOS-NWT [18], which are based on the Nonlinear Shallow Water Equations (NLSWE) and Potential Flow Theory (PFT) respectively, for the challenging problem of wave transformation due to nonlinear wave-wave interactions. The two numerical models of low computational cost are compared with experimental results and the CFD NWT designed in OpenFOAM, which demonstrated high accuracy for the present problem with limiting breaking [5] and breaking wave groups [19]. The two weakly nonlinear solvers can then be coupled with CFD, or simply be employed for preliminary studies and for performing the iterations of the focusing methodology for the correction of the input signal for the CFD. To confirm the applicability of the weakly nonlinear solvers for these purposes, the dispersion of steep waves and the induced nonlinearities should be performed to a similar level of accuracy as that of the CFD solver. The present work achieves this by effectively and consistently correcting the wave signal using the focusing methodology [12], and by examining in depth the wave propagation trough the comparison of the surface elevation of the wave group and the extracted harmonics at different locations in the flume. Such intercomparisons for wave propagation among NWTs are crucial also for the subsequent estimation of the wave loads and structural responses, as highlighted in the recent study of Ransley et al. [20]. To the best of the authors' knowledge, a cross-validation of the surface dynamics among the models OpenFOAM, SWASH and HOS-NWT has not been performed until now, and it contributes to the evaluating the capacity of the models for academic and operational applications.

The paper is organised as follows: Section 2 describes the experimental set-up and the focusing methodology. Section 3 introduces the numerical models OpenFOAM, SWASH and HOS-NWT, including details about the design of the NWTs and their convergence analyses. The numerical results are compared with the experimental results in Section 4 for the evolution of every harmonic of the wave group and their performance is discussed. A comparison between the extracted $2^{nd}$ order harmonics and the analytical $2^{nd}$ order theory solution is also included in the Appendix A. General conclusions and recommendations are drawn at the last section of the paper.

## 2. Methods and Testing Conditions

### 2.1. Experimental Conditions

The experiments were performed in the 20 m long flume at University College London (UCL). The flume was equipped with flap-type absorbing wavemakers, which operated with linear transport functions to produce a target spectrum or surface displacement at the desired location in the flume. The wave energy was dissipated by a parabolic beach at the opposite end of the flume to minimize any reflections. The selected working depth of the flume was 1 m. A schematic of the characteristics of the physical flume is depicted in Figure 1, including also the correction locations for the amplitudes and the phases of the extracted linear harmonics, AM and PF, respectively. The surface elevation was recorded with resistive wave gauges (WGs) at the locations mentioned in Table 1. The accuracy of the physical WGs was $\pm 1$ mm and their sampling frequency is 100 Hz.

A similar experimental set-up was used by Vyzikas et al. [5], but in the present study the accuracy is improved by repeating the experiments with only WG7 (PF) in the physical flume, to avoid even the marginally influences to the wave profile by the other intrusive physical WGs.

**Table 1.** Location of the wave gauges, as distance in *m* from the wavemaker (AM: amplitude matching; PF: phase focal).

| WG1 (AM) | WG2 | WG3 | WG4 | WG5 | WG6 | WG7 (PF) |
|---|---|---|---|---|---|---|
| 1.63 | 5.17 | 9.40 | 11.50 | 13.80 | 13.90 | 14.10 |

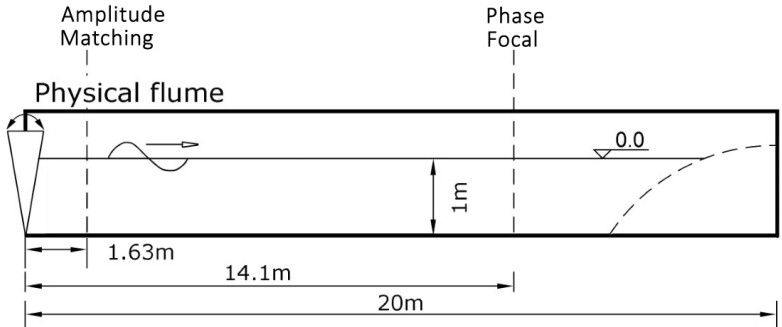

**Figure 1.** Schematic of the physical wave flume at UCL.

The NWTs, described in Section 3, are designed as numerical mirrors of the physical flume. There are however several differences that should be highlighted since they can explain the reasons for some of the discrepancies seen in the analysis of the results in Section 4. The main differences refer to (*a*) the wave generation, which is performed by a flap-type moving wave paddle in the laboratory and an inlet boundary condition in the numerical models, and (*b*) the absorption of the waves at the end of the flume, which is achieved by a parabolic beach in the physical flume and appropriate outlet boundary conditions with active or passive absorption in the NWTs. More specifically, in the NWT of OpenFOAM, the wave absorption is achieved locally at the outlet boundary, which is located at 20 m, corresponding to the end of the mechanical absorption at the physical flume, while in the NWTs of SWASH and HOS-NWT, the wave absorption is achieved by a sponge layer, extending the computational domains beyond 20 m. It should be noted here that neither the physical or numerical wave absorption methods can absorb 100% of the incident energy, resulting in reflections. In the NWTs, this can be partially attributed to the limitations of the numerical techniques used as well as the complex velocity field incorporated in a focused wave group, which is more complicated to handle than that of regular waves. Therefore, to mitigate this, the focusing location should be sufficiently far from the outlet boundary in order to avoid contamination of the measured signal with reflections, which then require more time to return from the outlet and be present at the focal location during the time window of interest.

Regarding the wavemaker conditions, in all flumes, linear wave generation is used, which for the case of the stationary numerical boundary takes the form of Equations (1) and (2) for the surface elevation and the velocities, respectively.

$$\eta = \sum_{i=1}^{N} \alpha_i cos(\kappa_i x - \omega_i t + \psi_i) \tag{1}$$

$$
\begin{aligned}
u &= \sum_{i=1}^{N} \alpha_i \omega_i \frac{cosh(\kappa_i z)}{sinh(\kappa_i d)} cos(\kappa_i x - \omega t + \psi_i), \\
w &= \sum_{i=1}^{N} \alpha_i \omega_i \frac{sinh(\kappa_i z)}{sinh(\kappa_i d)} sin(\kappa_i x - \omega_i t + \psi_i)
\end{aligned}
\tag{2}
$$

where $\eta$ is the free surface elevation; $u$ and $w$ the horizontal and vertical velocity components, (the normal to the NWT component $v = 0$); $\psi$ the phase of each wave component $i$; $z$ the distance from the bottom of the NWT; $x = 0$ m, horizontal distance from inlet boundary and $t$ the time.

The testing conditions refer to a broadbanded Gaussian amplitude spectrum. The spectrum is discretised by 320 equidistant frequency components ranging from 0.0078 Hz to 2.50 Hz, with a frequency increment $df = 0.0078$ Hz. Practically however, the components at frequencies higher than 1.5 Hz have zero amplitude. A large number of components guarantees adequate discretization of the spectrum, which can be crucial for the accurate reconstruction of the free water surface and the velocity profile at the inlet, which affect

the later dispersion of the wave group [11]. Moreover, for periodic focusing wave groups, the number of components and the repeat period (=$1/df$) should be such that zero surface elevation is achieved before and after the wave group and thus, the consecutive wave groups do not overlap and no wave-wave interactions occur [21]. Within the examined frequency range, the components near the spectral peak propagate in intermediate water depth, while the components at higher and lower frequencies propagate in deeper and shallower water regimes, respectively. This gives wider interest to the present study, since its validity is not constrained to very specific testing conditions.

The Gaussian shape of the amplitude spectrum is given by Equation (3), with standard deviation $\sigma = 0.13$ and peak frequency $f_p = 0.64$ Hz. In the present study, a strongly nonlinear limiting breaking wave group is examined, which has a linearly predicted amplitude of $A_{Th} = 0.154$ m. The wave conditions are listed in Table 2. Groups of lower steepness have been also tested, giving even better results, as expected.

$$E_a(f) = \frac{1}{\sigma\sqrt{2\pi}} e^{\left[\frac{-(f-f_p)^2}{2\sigma^2}\right]} \tag{3}$$

**Table 2.** Wave conditions.

| Gaussian Spectrum | |
| --- | --- |
| Peak frequency ($f_p$) | 0.64 Hz |
| Standard Deviation ($\sigma$) | 0.13 |
| $k_p d$ | 1.75 |
| Linear crest amplitude $A_{Th}$ (m) | 0.154 |

The reason for selecting a Gaussian spectrum as the target spectrum is its practical advantages compared with more realistic wave spectra, such as JONSWAP. For experiments, the full range of frequencies included in the spectrum can be efficiently generated by the physical wavemaker, while for spectra with a high frequency tail truncating the high frequency part at 2 or $3f_p$ is common practice and entails a rather sharp drop in the energy content of wave components with unknown consequences in the spectral evolution. In contrast, the selected Gaussian spectrum is the broadest possible, spanning smoothly in frequencies from 0 to $2f_p$, and, as the wave group propagates in the flume, a high frequency tail is developed, as demonstrated by Vyzikas et al. [5]. Additionally, a Gaussian spectrum has a compact shape of the timeseries of the free surface, consisting of one main crest and two deep troughs at focus, similar to PM spectra. On the other hand, the time history of the free surface for JONSWAP wave groups consists of many crests and troughs, being wider, and thus, requiring a longer flume for the simulation, to avoid reflections.

### 2.2. Phase Decomposition

The basic principle of the focusing methodology used in the present study is the correction of the extracted linear harmonics [12]. Thus, the success of the focusing methodology lies upon the accurate extraction of these harmonics, which can be achieved by means of the four-wave decomposition method. Some possible alternatives are discussed in this section to illustrate the relative advantages of the four-wave decomposition.

The phase or harmonic decomposition methods used in ocean and coastal engineering are based on the principle that the free surface elevation can be expressed by a Stokes expansion to power series, assuming that the $n^{th}$ order harmonic can be found from the envelope of the linear harmonics by raising the latter to the $n^{th}$ order [22]. To decompose the recorded wave signal to its harmonics, a number of wave groups with different phase shifts should be obtained, and their appropriate algebraic combination returns the harmonics of the signal. Generally speaking, a larger number of wave groups with different phases

shifts guarantee greater accuracy in the extraction of harmonics by reducing the need for frequency filtering to the signal.

The most widely used harmonic separation method is the two-phase decomposition, which requires a crest focused (CF) and a trough focused (TF) wave group to separate the signal in odd and even harmonics. The TF can be simply created by adding a phase shift of $\pi$ to the CF group, which is designed to have zero phases at focus. The extracted odd harmonics contain the linear, 3rd order and 5th order harmonics, while the extracted even harmonics contain the 2nd sum, 4th and 6th order harmonics. The linear harmonics are thus perturbed by the 3rd and the 5th order harmonics and cannot be readily isolated, unless the spectrum is sufficiently narrowbanded, resulting in no overlap between the linear and 3rd order harmonics. Under such conditions only, frequency filtering can be employed. However, this is rarely the case for realistic broadbanded spectra or for steep wave groups, whose free-wave spectrum may broaden considerably during focusing. As such, discrepancies are common in the literature, due to the limitation of the two-wave decomposition [22–25].

A more advanced method to extract the harmonics is the four-wave decomposition, as given in Equations (4). The separation is achieved by adding two extra shifts of $\pi/2$ and $3\pi/2$ to the two-wave decomposition method. The four-wave decomposition method was first suggested by Fitzgerald et al. [26] for forces on cylinders, in order to observe potential ringing effects. The method was extended to wave records by Stagonas et al. [12], following the same principle. Similar methods were used to study wave-structure interaction problems [27–29]. However, in those works, the decomposition was not combined with a focusing methodology and discrepancies in the results were observed.

The advantage of the four-wave decomposition method is that the linear and the 3rd order harmonics can be separated algebraically without the need for frequency filtering. Filtering is only needed to separate the linear from the 5th order terms, which is trivial, since these harmonics occupy distinctively different frequency bands. Moreover, the contribution of the 5th order harmonics is orders of magnitudes smaller than that of the linear and the 3rd order harmonics. The 2nd order super-harmonics, aka 2nd sum, can be readily extracted, while, for the 2nd order sub-harmonics, aka 2nd difference, trivial filtering from the 4th order harmonics is required.

Linear:
$$A f_{11} \cos \phi + A^3 f_{31} \cos \phi + O(A^5) = \frac{1}{4} \left( S_0 - S_{\pi/2}^H - S_\pi + S_{3\pi/2}^H \right) \tag{4a}$$

2nd sum:
$$A^2 f_{22} \cos 2\phi + A^4 f_{42} \cos 2\phi + O(A^6) = \frac{1}{4} \left( S_0 - S_{\pi/2} + S_\pi - S_{3\pi/2} \right) \tag{4b}$$

3rd:
$$A f_{33} \cos 3\phi + O(A^5) = \frac{1}{4} \left( S_0 + S_{\pi/2}^H - S_\pi - S_{3\pi/2}^H \right) \tag{4c}$$

2nd diff + 4th:
$$A^2 f_{20} + A^4 f_{44} \cos 4\phi + O(A^6) = \frac{1}{4} \left( S_0 + S_{\pi/2} + S_\pi + S_{3\pi/2} \right) \tag{4d}$$

where $f_{ij}$ the coefficients in Fourier series, $A$ the amplitude of the envelope and $S_{n\pi/2}$, $n = 0, 1, 2, 3$ the timeseries of the surface elevation at the location of interest. The superscript $H$ refers to the imaginary part of the conjugate of the Hilbert transform of the corresponding timeseries of the surface elevation.

It is noted that the relative importance of the term $f_{31}$ is small compared to the term $f_{11}$, because, despite of having the same frequency dependence, its amplitude dependence is at high order, and thus, it is considerably smaller than $f_{11}$. Similar is the case for the terms $f_{42}$ and $f_{22}$. In general, this holds for all the difference terms $f_{ij}$, $i \neq j$, instead of the 2nd difference terms $f_{20}$, which are important and they can be separated by frequency filtering from the 4th order terms ($f_{44}$) [26]. Here, the cut-off frequency for the 2nd difference harmonics is approximately taken at $2.5 f_p$.

A more accurate separation of harmonics can be achieved by the twelve-wave decomposition of Hann et al. [30], who considered phase shifts of an increment of $\pi/6$. This method practically eliminates the need for filtering for the separation of harmonics.

As such, the 4th order harmonics are readily separated from the 2nd difference harmonics and the 5th order harmonics are calculated separately. Nevertheless, for the wave group examined by Hann et al. [30], the performance of the twelve-wave decomposition was similar to the four-wave decomposition regarding the extracted surface elevation. Therefore, the use of the twelve-wave decomposition is not deemed necessary, especially because the simulation of eight extra groups at every iteration step of the focusing methodology requires considerable additional computational resources.

*2.3. Correction Methodology*

The methodology for focusing wave groups used in the present study was proposed by Stagonas et al. [12] and applied in a number of studies, such as in Vyzikas et al. [31] (first time application in a numerical model), Vyzikas et al. [32] (comparison between CFD and NLSWE), Buldakov et al. [33] (Lagrangian solver including sheared currents), Buldakov et al. [34] (breaking waves), Vyzikas et al. [5] (evolution of harmonics) and Stagonas et al. [19] (breaking waves in CFD). More recently, the methodology was compared with other focusing methods, showing the best performance [35].

The methodology is an iterative correction process of the amplitudes and phases of the wave components of the spectrum until they match the desired target values. The corrections of the amplitudes and phases can be performed at different locations depending on the scope of the study, allowing for examination of specific properties of the wave group, e.g., dispersion [5]. The distinct characteristic of the methodology is that the corrections of phases and amplitudes are performed only for the components of the extracted free-wave spectrum. Therefore, high accuracy in extracting the linear harmonics is needed, which is achieved by combining the focusing method with a four-wave decomposition technique, elaborated in Section 2.2.

The steps for applying the focusing methodology are:

1. The target amplitude spectrum is defined and the desired locations for the amplitude and phase corrections, namely AM and PF, respectively, are determined. Moreover, the focal time is selected, usually as half of the repeat time of the periodic signal.
2. Wave groups of different phase shifts are generated at the wave paddle. For a four-wave decomposition, four wave groups with phase shifts of $0$, $\pi/2$, $\pi$ and $3\pi/2$ are used to generate CF, positive slope, TF and negative slope focused waves at the PF location, respectively. For the first run, the linear dispersion relation can be used to backwards propagate the signal from PF to the wavemaker, as a best guess. An example is given in Figure 2, where the contraction of the wave group towards focusing is also evident.
3. The linear harmonics are extracted using a suitable linear combination of the four wave groups measured at PF, according to the four-wave decomposition (see Equation (4)) in the frequency domain after performing a Fast Fourier Transform (FFT) of the measured signals.
4. The phases and amplitudes of the wave components of the linear harmonic are corrected using Equations (5).

$$\alpha_{in}^{i+1} = \alpha_{in}^{i} \times \alpha_{trg}/\alpha_{out}^{i} \quad \text{and} \quad \phi_{in}^{i+1} = \phi_{in}^{i} - (\phi_{trg} - \phi_{out}^{i}) \tag{5}$$

where $\alpha_{in}, \alpha_{out}, \alpha_{trg}$ are the input, measured and target amplitudes of the components of the linear spectrum respectively and $\phi_{in}, \phi_{out}, \phi_{trg}$ are input, measured and target phases of the components of the linear spectrum, respectively.
5. The corrected signal for the wavemaker can then be calculated: (*a*) the phases of wave components of the corrected linear spectrum are found by propagating backwards the signal from PF to the wavemaker using the linear dispersion relation, while (*b*) the corrected amplitudes of the components are not altered according to linear theory, being the same at AM and the wavemaker.

6. The process is repeated iteratively from step 2 to 5 until the target values for $\alpha$ and $\phi$ match the target values within the desired accuracy.

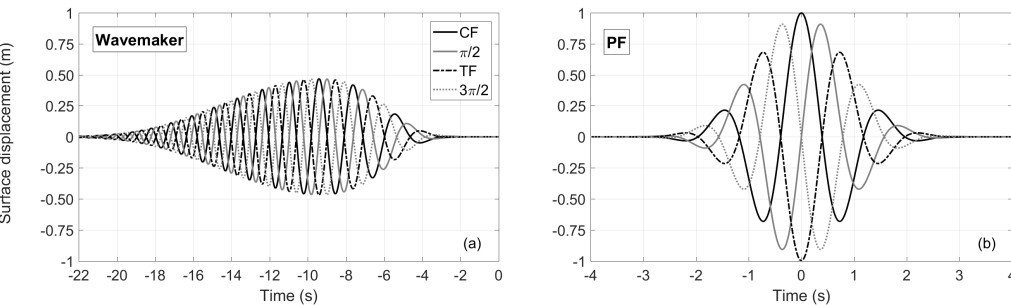

**Figure 2.** Timeseries of the normalized free surface elevation of four wave groups of different phases at the wavemaker (**a**) and at the PF location (**b**) calculated by linear theory.

The distinguishing characteristic of the validation presented in the present study is that the amplitude spectrum is corrected at a different location (AM) than the phases of the wave components (PF). The AM location is selected to be close to the wave paddle, while the PF point is located at a downstream location. Therefore, the correction methodology is performed in two steps, independently for the phases and the linear amplitude spectrum. The advantages of this approach are both practical and theoretical. By correcting the spectrum not at the inlet, but downstream in the wave tank, the discrepancies induced by the wave paddle are eliminated and the target spectrum is reproduced exactly in the nonlinear domain or physical wave tank. Moreover, the wave group is dispersed and less steep near the wavemaker, and thus closer to the linear solution, reducing the challenges in the correction of the extracted linear spectrum. At the same time, this method also enables the examination of the natural evolution of the amplitude spectrum from the AM location, where the wave group is dispersed, until the PF, where the strongest wave-wave interactions occur as the wave group takes its steepest form. Therefore, instead of forcing the wave group to take the target spectral shape at the focal point, the wave group has the target spectral shape close to the inlet and it is allowed to evolve freely from the AM to PF location according to the "dispersive" properties of the NWTs, which in physical terms are controlled by the third-order wave-wave interactions. It should be noted that although the focusing methodology forces the solution for the phases of the linear harmonics at the PF, it does not directly pre-determine the magnitude of the nonlinear harmonics, which affect the shape and the height of the resulted focused wave group. These nonlinear harmonics, together with the evolution of the underlying linear spectrum, depend on the capacities of the NWTs in simulating nonlinear waves and do not constitute artefacts of the iterative correction process of the focusing methodology.

## 3. The Numerical Models

### 3.1. RANS: OpenFOAM

3.1.1. Description of the Solver

OpenFOAM (Open source Field Operation and Manipulation) is an extensive software package for solving continuous mechanics problems. It was initially developed in the late 1980s at Imperial College, London and it was later rewritten in C++, incorporating the advantages of object oriented programming [36,37]. OpenFOAM releases are open-source and freely available under GNU General Public Licence and, nowadays, it is widely used for industrial and academic CFD applications [38]. The version used in the present study is 2.1.x.

Regarding free surface fluid flows, OpenFOAM can solve the 3-Dimensional (3D) NSE multiphase flows including turbulence, using the Finite Volume Method (FVM) discretization with the Volume of Fluid Method (VoF) [39]. This allows for employing the model for

nonlinear problems with highly distorted free surface, such as wave breaking and interaction of waves and structures. The code has been applied for coastal and ocean engineering problems after the development of appropriate boundary conditions for wave generation and absorption, mainly through the libraries IHFOAM [40] and waves2Foam [41]. Here, the former is used, thanks to its higher computational efficiency, as discussed in [5]. Both these integrated libraries employ the "interFoam" solver for two-phase incompressible NSE. In the present study, the Reynolds Averaged Navier-Stokes (RANS) approach is employed as a common and relatively efficient way to solve the NSE [42]. The continuity (Equation (6)) and momentum (Equation (7)) equations are solved simultaneously for the two Newtonian and immiscible fluids (air and water).

$$\nabla \mathbf{U} = 0, \tag{6}$$

$$\frac{\partial \rho \mathbf{U}}{\partial t} + \nabla \cdot (\rho \mathbf{U} \mathbf{U}) - \nabla \cdot (\mu_{eff} \nabla \mathbf{U}) = -\nabla p^* - g \cdot \mathbf{X} \nabla \rho + \nabla \mathbf{U} \cdot \nabla \mu_{eff} + \sigma_\tau \kappa_c \nabla \gamma_i \tag{7}$$

where $\mathbf{U}$ is the velocity vector, $\rho$ is the density, $p^*$ the pseudo-dynamic pressure, $\mathbf{X}$ the position vector, $\sigma_\tau$ the surface tension coefficient (0.07 Kg/s$^2$), $\kappa_c$ the curvature of the interface, $\gamma_i$ the fluid phase fraction and $\mu_{eff}$ the efficient dynamic viscosity. $\mu_{eff} = \mu + \mu_t$, with $\mu$ being the molecular dynamic viscosity ($10^{-3}$ m$^2$/s and $1.48 \times 10^{-5}$ m$^2$/s for water and air, respectively) and $\mu_t$ is the turbulent viscosity given by the turbulence model [43]. Here, since no breaking waves are simulated, a quasi-laminar flow model is employed.

### 3.1.2. Design of the NWT

The NWT is designed as a two-dimensional (2D) numerical mirror of the physical wave flume at UCL, seen in Figure 1, having a length of 20 m. The computational mesh has a similar design to that presented in [5], consisting of three layers: a middle layer of square cells (aspect ratio, AR = 1), which has the highest resolution and encapsulates the maximum and minimum free surface extending $\pm 0.2$ m from the still water level (SWL); a top layer 0.2 m wide of maximum cell AR = 2 extending to the atmospheric boundary; a lower layer 0.8 m wide of maximum cell AR = 4 extending to the bottom of the NWT. The refinement around the interface of the two fluids is performed with the utility "snappyHexMesh" [44,45].

The computational domain is a closed one-cell thick rectangular consisting of six walls with assigned appropriate boundary conditions for every variable as listed in Table 3, and elaborated in [5]. "Empty" is used for transforming a 3D domain to a 2D one, here used to reduce the computational effort. To further reduce the computational cost only a short part of the timeseries is simulated, between times 40 s and 70 s, with the focusing event being at 64 s.

**Table 3.** Boundary condition for the NWT in IHFOAM [45].

| Boundary | $\gamma_i$ | *Pressure* | *Velocity* |
|---|---|---|---|
| Inlet | IH_Waves_InletAlpha | buoyantPressure | IH_Waves_InletVelocity |
| Outlet | zeroGradient | buoyantPressure | IH_3D_2DAbsorbtion_InletVelocity |
| Top | inletOutlet | totalPressure | presureInletOutletVelocity |
| Bottom | zeroGradient | buoyantPressure | fixedValue |
| Lateral walls | empty | empty | empty |

The time stepping is controlled by the Courant condition ($C_o$) [46], which represents the portion of the cell that the advective flow can cover in one time-step. An additional time controller ($alphaC_o$) for the interface of multiphase flows is also used to ensure stability. OpenFOAM also includes various numerical schemes for the spatial and temporal discretization of the partial differential equations, which were selected based on preliminary investigations [31], aiming for maximum accuracy and optimal computational cost.

### 3.1.3. Convergence Tests

On achieving grid-independent solution, focused waves were generated for combinations of a R2.5-C0.1, R2.5-C0.2, R5.0-C0.2 and R10-C0.2, where R is the minimum cell size in mm and C is the value of $C_o$, which was selected to be the same as $alphaC_o$. The results of the convergence analysis for the measured surface elevation are shown in Figure 3 for the steepest wave group at the PF location. It is noted that for each R-C combination, the wave group was accurately focused using the focusing methodology in order to minimise the additional influence of the occurring wave-wave interactions that affect the dispersion of the wave group. Vyzikas et al. [5] presented a more in-depth analysis including the individual harmonics, showing that coarser resolutions and higher $C_o$ result in significant overestimations of the 3rd + higher order harmonics and deeper 2nd order wave troughs when the focusing methodology is employed. Here, it is shown that the overall increase of the crest amplitude for the coarser resolutions reaches 20%. Coarser resolutions than the R10-C0.2 cause local distortions of the crest, which can be associated with premature breaking. The total number of cells in the computational domain is 2.48 millions for the high resolution NWT that was finally chosen (R2.5-C0.1), resulting to a computational time of approximately 50 h on a 16-core Intel Xeon E5-2650 @ 2.6 GHz.

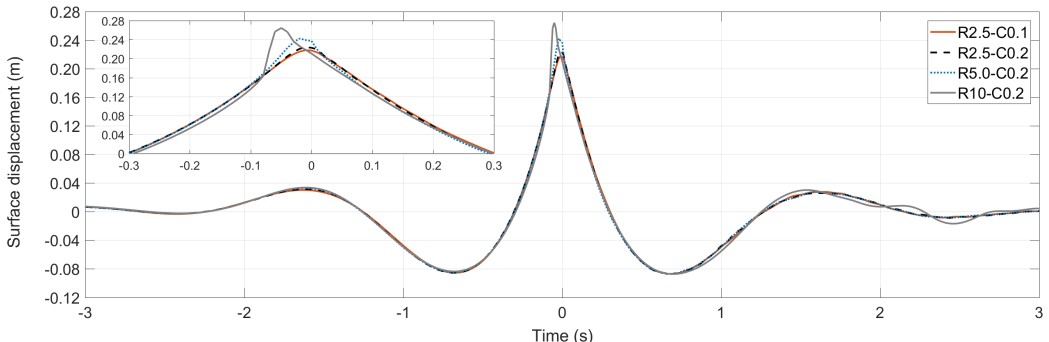

**Figure 3.** Convergence analysis for the NWT in OpenFOAM.

### *3.2. NLSWE: SWASH*

#### 3.2.1. Description of the Solver

SWASH (Simulating WAves till SHore) is a general-purpose numerical model for rapidly changing flows at arbitrary water depth, which is based on the NLSWE with a non-hydrostatic pressure assumption. SWASH is an open-source freely distributed model that was developed in TU Delft by Zijlema et al. [17] following precious works [47–50]. The code is written in FORTRAN and its structure resembles the well-established spectral model SWAN. SWASH is considered an operational model for coastal engineering problems at regional scale, but thanks to its flexibility, it can be applied to different space and time scales ranging from field to laboratory studies. In the present work, version 3.14 was used, which had already incorporated all the physics and numerical techniques of the most recent versions, regarding the propagation of steep non-breaking waves over constant depth.

The governing equations of the depth-averaged non-hydrostatic free surface flow described by the non-conservative form of the NLSWE are presented in Cartesian notation in Equations (8) and (9). These equations can be derived from the two-dimensional form [17] after some substitutions by considering only the vertical plane ($y = 0$), zero bottom friction ($c_f = 0$) and no turbulent stresses ($\tau_{ij} = 0$), which are not relevant for the present study. Bottom friction becomes important only for rough bathymetry and when waves travel for long distances, which are not the case for focused waves over flat bottom. Moreover, eddy viscosity is also considered negligible ($v_t = 0$ and thus $\tau_{ij} = 0$), since no breaking is encountered in the present tests and the waves do not propagate over strong sheared currents [51].

$$\frac{\partial \zeta}{\partial t} + \frac{\partial hu}{\partial x} = 0 \tag{8}$$

$$\frac{\partial u}{\partial t} + u\frac{\partial u}{\partial x} + g\frac{\partial \zeta}{\partial x} + \frac{1}{2}\frac{\partial q_b}{\partial x} + \frac{1}{2}\frac{q_b}{h}\frac{\partial(\zeta - d)}{\partial x} = 0 \qquad (9)$$

where $t$ is time, $x$ and $z$ located at the still water level (SWL) and the $z-$axis pointing upwards, $\zeta(x,t)$ is the surface elevation measured from the SWL, $d(x)$ is the local water depth from the SWL, $u(x,t)$ is the depth averaged flow velocity in $x$-direction, $g$ is the gravitational acceleration, $q(x,z,t)$ is the non-hydrostatic pressure (normalised by the density), which is calculated here from the non-hydrostatic pressure at the bottom $q_b$. $h = \zeta + d$ is the instantaneous water depth or total depth.

The numerical implementation of the NLSWE is performed on a staggered grid for the calculation of the flow variables based on the FVM in a domain bounded by the sea bed and the free surface. The computational efficiency of SWASH lies on the $\sigma$-transformed vertical grid, which forms layers of varying thickness, and on the Keller-box method, which allows for accurate calculation of the pressure at the free water surface even with a small number of vertical layers. A semi-implicit time integration is used to solve the incompressible NSE, averaged per layer, aiming for both numerical robustness as well as a good balance between computational efficiency and accuracy [49]. Regarding overturning waves, SWASH considers a single-value free surface and a bore analogy, resembling a moving hydraulic jump with energy dissipation. However, this feature is not activated in the present work, since no breaking waves are considered.

### 3.2.2. Design of the NWT

The NWT in SWASH is designed similarly to the NWT in OpenFOAM (see Section 3.1.2). The main difference is that the computation domain is extended to 30 m in order to accommodate a sponge layer of 10 m for the wave absorption at the outlet.

The wave generation in SWASH is performed with the "FOURIER" inlet boundary condition, referring to a stationary boundary that calculates the surface elevation and the velocity for every layer. For the simulation of irregular waves, a weakly reflective boundary condition is also used at the inlet boundary to minimize any returning reflections or potential local disturbances.

### 3.2.3. Convergence Tests

A convergence analysis for SWASH was performed by Vyzikas et al. [32] for a similar problem of focusing wave groups of moderate steepness. In that study, a thorough analysis of more than 40 combinations of the parameters including numerical schemes and resolutions was tested aiming to balance accuracy and computational efficiency. The selected computational grid consisted of 6 layers of variable thickness of 5%, 10%, 15%, 20%, 25% and 25% of the water depth calculated from the free surface to the sea bed, with finer layers towards the free water surface. The horizontal grid was uniform with a cell size of 40 mm. Regarding the numerical schemes employed, the Keller-box scheme was used for the calculation of the vertical pressure gradient using an implicit Euler scheme. A second order backward difference (BDF) numerical scheme was used for the discretization of the momentum and transport equations, as well as the water depth in velocity points. A semi-implicit Crank-Nicolson scheme was used for the time integration of the continuity equation and the water level gradient, allowing the Courant number to take values greater than 1 and the stability not to depend only on the long wave celerity, which would slow down the computation. A high relative accuracy of the solvers (0.001) and a high number of maximum iterations (1000) were chosen for achieving accurate results. The ILU preconditioner was used for the calculation of the non-hydrostatic pressure. A uniform distribution of the velocity was considered per layer.

Although the aforementioned set-up for the SWASH NWT has much higher resolution than those commonly used for engineering applications (1–2 layers), here a further refinement is performed as a further convergence check to verify the accuracy of the NWT. The number of layers is increased to 8 layers of relative thickness of 2%, 4%, 9%, 15%, 15%, 20%, 20%. Additionally, a hyperbolic distribution for the horizontal velocity per layer is

considered, which should be more accurate for the high frequency wave components that propagate in deeper water.

The convergence check for SWASH is presented in Figure 4. The difference of the measured surface elevation at the crest is only 1 mm with the higher resolution giving a higher crest elevation. On the other hand, the time history of the surface elevation shows that the shape of the wave group is practically identical. For the validation of SWASH presented in Section 4, the high resolution NWT is selected, inducing a moderate increase of the computational cost of about 30%, requiring approximately 20 min on a single core of 2 GHz for 30 s of simulation.

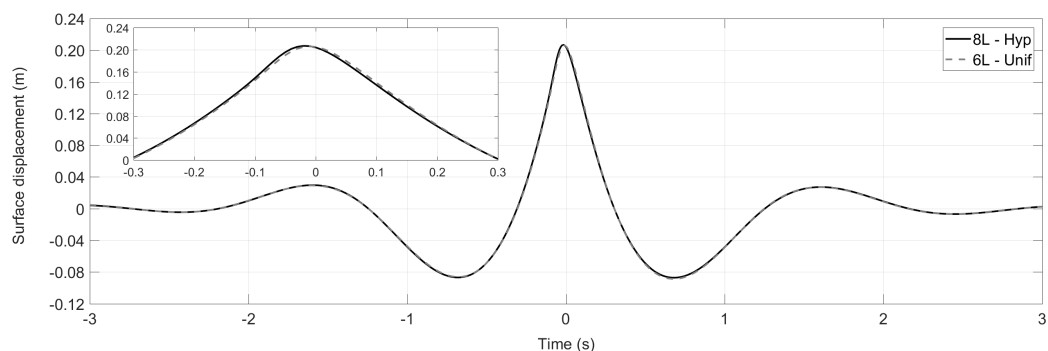

**Figure 4.** Convergence analysis for the NWT in SWASH.

*3.3. PFT: HOS-NWT*

3.3.1. Description of the Solver

The PFT model used in the present study is the open source HOS-NWT that was developed at the LHEEA laboratory of Ecole Centrale de Nantes (ECN). There are two version of HOS solvers, namely HOS-NWT [18] and HOS-ocean [52], which follow the earlier developments of SWEET (spectral wave evolution in the ECN tank) [53], SWENSE (Spectral Wave Explicit Navier-Stokes Equations) and HOST (High Order Spectral method Tank) [54]. HOS-ocean and HOS-NWT models share the same equations and solving techniques and essentially, they only differ at the treatment of the boundary conditions: HOS-ocean has periodic boundaries representing an infinite circulating ocean, while HOS-NWT has a bounded domain with wave generation and absorption boundary conditions and reflective walls elsewhere. The fundamental assumption of the PFT is the consideration of an irrotational, inviscid, incompressible fluid allowing for the continuity equation to be expressed in the form of Laplace equation for the velocity potential $\phi$, as seen in Equation (10) [52]. This method is based on the early works of Dommermuth and Yue [55] and West and Brueckner [56].

$$\nabla^2 \phi + \frac{\partial^2 \phi}{\partial z^2} = 0 \tag{10}$$

where $\nabla$ is the horizontal gradient operator $(\partial_x, \partial_y)$ and $z = 0$ is located at the mean water level (MWL).

Since non-breaking waves are considered, the free surface is a single-valued variable at any location in the domain. Next, the kinematic and dynamic boundary conditions at the free surface (Equations (11) and (12)) should be defined in order to close the system of equations. For this, the surface quantities are used, i.e., the free surface elevation $\eta$ and the free surface velocity potential $\tilde{\phi}(x, t) = \phi(x, z = \eta(x, t), t)$, according to the formulation of [57].

$$\frac{\partial \eta}{\partial t} = \left(1 + |\nabla \eta|^2\right) W - \nabla \tilde{\phi} \cdot \nabla \eta \tag{11}$$

$$\frac{\partial \tilde{\phi}}{\partial t} = -g\eta - \frac{1}{2}|\nabla \tilde{\phi}|^2 + \frac{1}{2}\left(1 + |\nabla \eta|^2\right) W^2 \tag{12}$$

where $W$ refers to the vertical velocity at the free surface $W = \frac{\partial \phi}{\partial z}(x, z = \eta, t)$ as expressed by [56].

The HOS-NWT is a very computationally efficient solver for waves, thanks to the fact that it exploits the natural representation of the waves in a Fourier spectrum. The computational cost of HOS-NWT can be approximately 10–60 times lower compared to SWASH, which is already orders of magnitude faster than OpenFOAM. On the other hand, the price for this is the heavily parameterized wave breaking and the intrinsic limitations in simulating submerged (or surface piercing) structures. Regarding the simulation of extreme waves in HOS, the method is ideal to study "naturally" emerging extreme waves in random sea simulations thanks to its low computational cost [58]. In previous studies, HOS-NWT was also used to compare experimental results for 3D focused wave group of low steepness and a unidirectional group of moderate steepness [18]. However, to the best of the authors' knowledge, HOS-NWT has not been validated against experimental results for steep focused wave groups and the propagation of individual harmonics prior to the present study.

### 3.3.2. Design of the NWT

The NWT in HOS-NWT has a similar design to that of SWASH and OpenFOAM. For wave generation, HOS-NWT offers the possibility to simulate different types of wavemakers of linear or higher order motion. In the present study, a linear piston wavemaker starting at the bottom of the flume was selected with a linear ramp up function of 5 s, to allow for smooth starting of the simulation. The modelling of the wavemaker requires special treatment in HOS method. For this, HOS-NWT follows the same principle as in SWEET [53], with the wavemaker having continuous geometry and a no-flow condition. Moreover, to accurately model the movement of the wavemaker without increasing the energy and volume of fluid in the domain, the concept of the additional potential, coined by Bonnefoy et al. [53] following previous works [59], is employed.

The length of the NWT is 50 m and the depth 1 m. An absorption zone ("numerical beach") is employed for the dissipation of the incident waves at the end of the NWT. It starts at 40 m and occupies 20% of the numerical domain. Different lengths of the NWT were initially tested [60], in order to examine any potential reflections at the PF location, but the aforementioned set-up was deemed sufficient. The efficiency of the numerical beach is also controlled via the weighting function of the relaxation. Here, the default third order polynomial function is chosen [18] and a Runge-Kutta 4$^{th}$ order scheme for the integration in time, with a tolerance at $10^{-4}$ is selected after tests. The input signal for the simulation is given by an amplitude-frequency spectrum, using $icase = 3$ of HOS-NWT as the initial point for the present set-up.

### 3.3.3. Convergence Tests

The main parameters in the HOS-NWT are the number of nodes/modes in $x$-direction ($N1$), in $y$-direction ($N2$), at the wavemaker ($N3$), the dealiasing in $x$- and $y$-direction ($p1$ and $p2$ respectively) and the HOS nonlinearity order ($mHOS$). For a 2D simulation of unidirectional wave groups, $N2 = p2 = 1$. As common practice suggests, full dealiasing is used by defining $p1 = mHOS$.

The convergence of the HOS-NWT is performed by examining the three main parameters of the model, i.e., $N1$, $N3$ and $mHOS$. The convergence analysis is planned according to the following strategy: for a high order of HOS, i.e., $mHOS = 8$, and the same $N3 = 33$, the value of $N1$ is examined. Then, for $mHOS = 8$ and the highest $N1$ ($N1 = 1025$), the influence of $N3$ is examined. At the end, after selecting $N1$ and $N2$, the order of HOS is examined. This analysis is summarized in Figure 5. It can be seen that small discrepancies are observed for the lowest value of $N1$, but the agreement between $N1 = 513$ and $N1 = 1025$ is excellent. Thus, a value of $N1$ close to 513 is selected. The increase of $N3$ does not improve the results of the simulation and thus, any value of $N3$ higher than $N3 = 33$ is sufficient. Finally, the order of HOS does not cause any noticeable improvement on the

results. *mHOS*, however, is the parameter that increases the most the computational cost and thus, the lowest possible value was considered, retaining a fully nonlinear simulation. For the results presented hereafter, the selected values of the converged HOS-NWT are: $mHOS = 5$, $N1 = 500$ and $N3 = 40$.

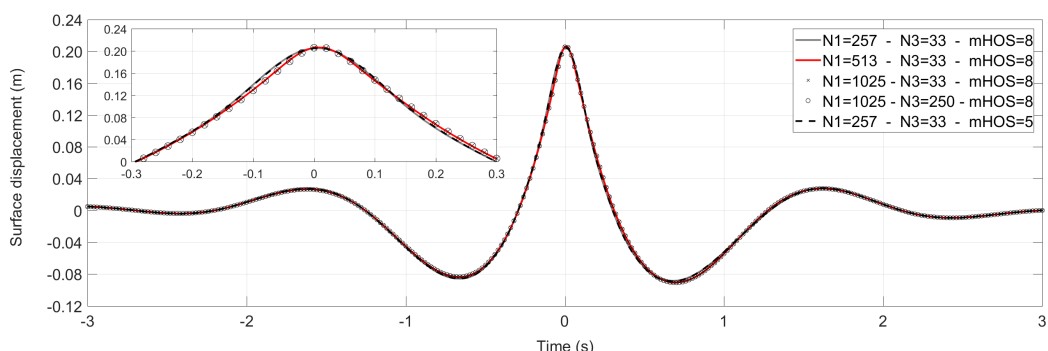

**Figure 5.** Convergence analysis for the HOS-NWT.

### 3.4. Summary of NWTs

The characteristics of the NWTs discussed in the previous sections are summarized in Table 4. It is noted that the computational cost is estimated approximately, because the simulations were performed at different platforms.

**Table 4.** Summary of the characteristics of the NWTs.

| NWT Parameters | OpenFOAM | SWASH | HOS-NWT |
|---|---|---|---|
| Equations | RANS | NLSWE | PFT |
| Mesh | quasi 3D static | 2D moving | 1D spectral |
| Wave generation | vel. distribution | vel. distribution | piston |
| Wavemaker motion | stationary | stationary | moving |
| Wave absorption | active | passive | passive |
| Length of NWT | 20 m | 30 m | 50 m |
| No. cells/nodes | $2.48 \times 10^6$ | $6 \times 10^3$ | $0.5 \times 10^3$ |
| Comp. cost (core hours) | 800 | 0.4 | 0.05 |

## 4. Results and Comparisons

In this section, the numerical models are validated against the experimental results and compared all together by examining the evolution of the nearly breaking wave group from the AM to the PF locations. First, the wave group dispersion is studied in the spectral domain and later the evolution of the individual harmonics at consecutive locations is examined in the time domain. In this way, conclusions can be drawn for both the numerical dispersion of the models and the emerging nonlinear wave-wave interactions. At the end of the section, a more detailed quantitative comparison is performed at the PF location for the extracted harmonics up to 5th order.

### 4.1. Numerical Dispersion

The backbone of the present analysis is the methodology for focusing waves, which effectively corrects the particularities of each flume and allows for obtaining the target phases and amplitudes at the desired locations. Here, the amplitudes of the extracted linear wave components are corrected close to the paddle at the AM location and they are let to evolve freely downstream. Similarly, the phases of the extracted linear wave components are only corrected at the PF location, being able to observe how they evolve freely from upstream. Thus, for verifying the numerical dispersion of the NWTs, the phases and amplitudes of the extracted linear wave components should be examined from AM to PF and from PF to AM, respectively. This comparisons are performed in the present section, including an intermediate location at 9.4 m (WG3) from the wavemaker.

The evolution of the amplitude spectra of the extracted linear harmonics is presented in Figure 6. It can be seen that at AM the spectra match the target amplitude spectrum, with some minor discrepancies appearing only at the spectral peak, where OpenFOAM and SWASH give a marginal overestimation, and at low frequencies, where OpenFOAM induces marginal increase. As the wave group propagates, the extracted linear spectra are downshifted and an energy increase at high frequencies is observed compared to the original target spectrum. The best agreement among all the models and the experiment is observed at WG3 (Figure 6b), possibly because the wave group is far from the inlet boundary but still not at its steepest form. At PF, the agreement among the models is very good, with OpenFOAM giving again a marginal increase near the spectral peak. However, as seen in Figure 6c, the experimental results have local discrepancies, which can be attributed to the beginning of breaking.

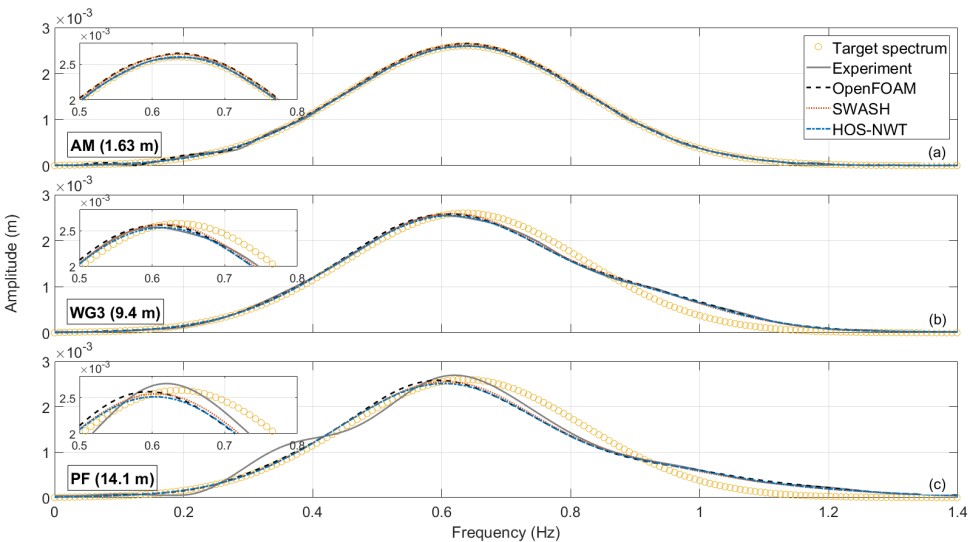

**Figure 6.** Amplitude dispersion from (**a**) AM, (**b**) WG3, (**c**) PF of the extracted linear harmonics.

The dispersion of the phases of the extracted linear wave components is presented in Figure 7. To facilitate comparisons, instead of presenting the raw phases ($\phi$) from the FFT of the extracted linear harmonics, the $\sin(\phi - \kappa x)$ is plotted, where $\kappa$ is the wavenumber and $x$ the distance from the PF. It can be seen that at PF (Figure 7c) the linear wave components are in phase. Discrepancies are manifested only at low and high frequencies, where, however, the energy content is very small, as seen from the spectra in Figure 6. When the wave group is at its more dispersed stage (AM, Figure 7a), the best agreement is observed between OpenFOAM and HOS-NWT, while SWASH has more similar results to the experiment. The similarity of the phases between OpenFOAM and HOS-NWT may indicate good potential for integration of the models.

The results in this section demonstrated that all the numerical models can capture the changes dispersive properties of the extracted free waves of the group to a very good extent, especially for the part of the spectrum with appreciable energy content (0.2–1.2 Hz). In the next sections, the comparisons will be performed in the time domain in order to identify the combined effects of the amplitudes and phases of the wave components.

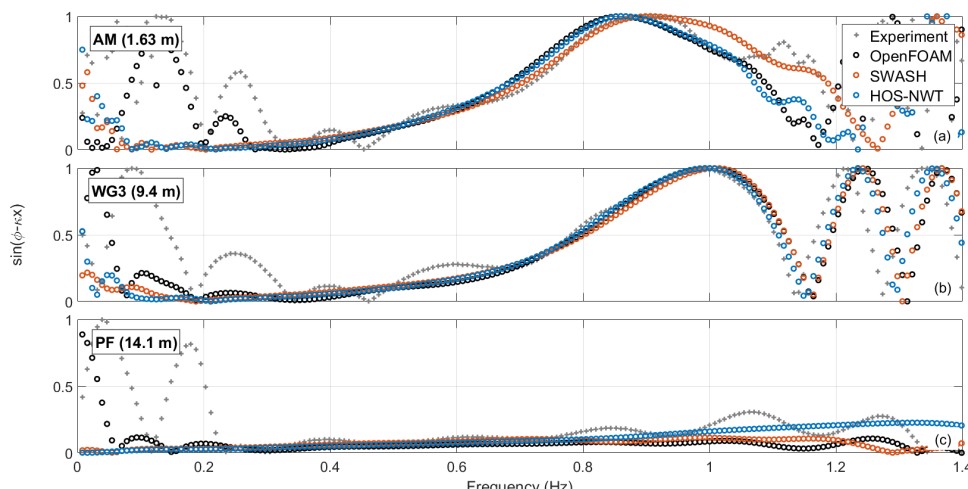

**Figure 7.** Phases dispersion from AM (**a**), WG3 (**b**), PF (**c**) of the extracted linear harmonics.

### 4.2. Evolution of the Linear Harmonics

The reproduction of the evolution of the linear harmonics is crucial for the accurate overall evolution of the wave group, because on one hand, the linear harmonics contribute the most to the measured surface elevation (up to 75% [5]), and on the other hand, they correspond to the free-wave components of the wave group, and thus, they control the evolution of the bound harmonics. Figure 8 demonstrates that the agreement of the extracted linear harmonics of all the models is almost excellent at all the WGs. The numerical models are in almost absolute agreement among each other, showing both that the focusing methodology is effective in every model, and that all the models are capable of dispersing very steep waves groups. Minor discrepancies are only observed for the experimental results, before and after the main wave group, possibly caused by the spurious effects of the wave generation and imperfect wave absorption. These may also be connected to the differences at the amplitude spectrum of the experimental linear harmonics in Figure 6c.

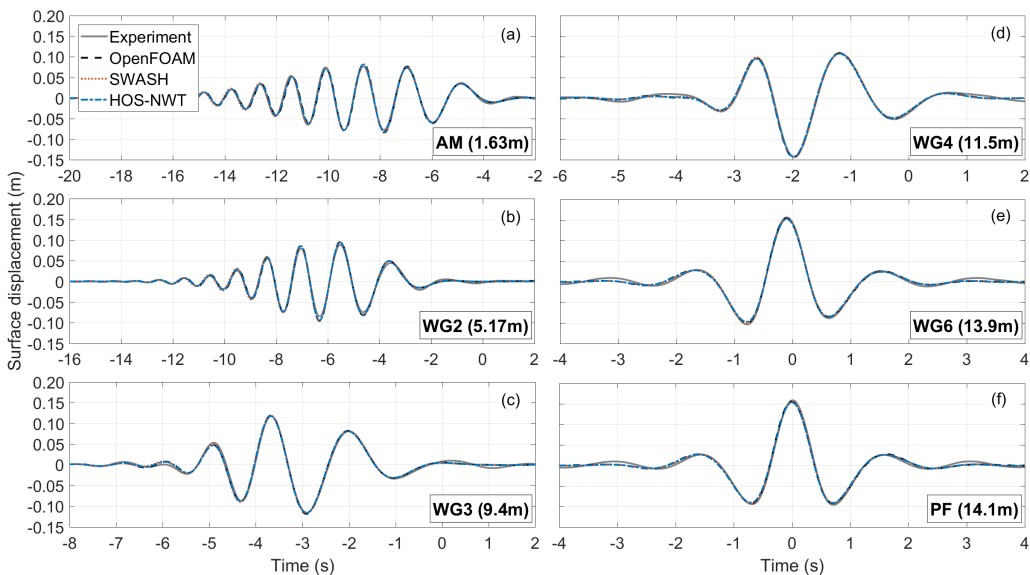

**Figure 8.** Comparison of the free surface elevation of the linear harmonics between the numerical models and the experiment at different locations (**a–f**).

### 4.3. Evolution of the 2nd Sum Harmonics

The nonlinear harmonics examined in the present and following sections emerge naturally from the linear harmonics as bound waves due to the high steepness of the group, and as spurious free waves caused by deficiencies of the wave generation. The comparison between the numerical models and the experiment for the extracted 2nd order sum harmonic is presented in Figure 9. It can be seen that the numerical models produce practically identical results everywhere in the NWT. Discrepancies exist only near the wavemaker at WG2 due to the spurious free waves created by the linear wave generation shown between −3 s and 1 s. Although, all wavemakers operate linearly, their function is not identical, especially when compared to the physical motion of the experimental wave paddle, which introduces greater spurious waves. From WG3 and downstream the agreement between the models and the experiment is almost excellent, since the spurious 2nd order free waves have separated from the wave group.

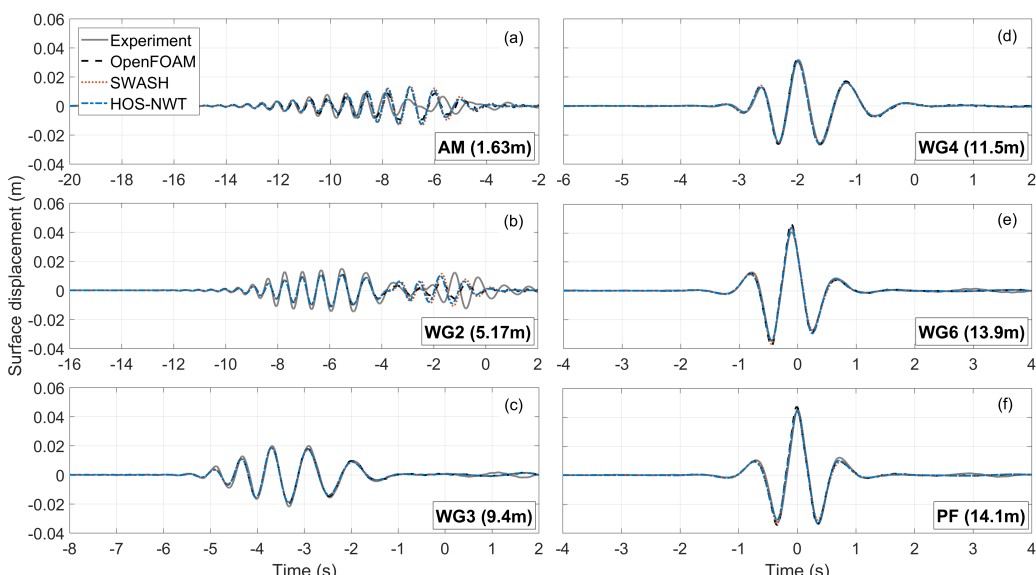

**Figure 9.** Comparison of the free surface elevation of the 2nd order sum harmonics between the numerical models and the experiment at different locations (**a**–**f**).

### 4.4. Evolution of the 2nd Difference Harmonics

The 2nd order difference harmonics refer to the long bound wave components that appear in the form of a set-down of the MWL under unidirectional wave groups. However, in directional seas, the long bound wave can take also the form of a set-up, for example when two wave groups cross at a specific angle, as discussed for the Draupner wave [61]. The results in Figure 10 show that there can be non-negligible discrepancies among the models and the experiments at the reproduction of the 2nd order difference harmonics.

In more detail, at the AM location (Figure 10a), it is seen that the 2nd order difference harmonics are practically zero for the experiment, SWASH and HOS-NWT, while OpenFOAM gives a spurious set-up before the main set-down of the wave group, which, precedes the main wave group [5]. Also, the experimental results show an artificial elevation at WG2, which appears to be a spurious local effect. From WG3 and downstream, the agreement between the models and the experiment improves considerably, especially for reproducing the main trough. SWASH has the shallowest trough at almost all locations, but at PF shows the best agreement with the experiment. At the last two WGs, the experimental results show a second trough at later times, which is the reflected long wave. At the examined time window, reflections do not appear in the SWASH and HOS-NWT, because the outlet boundary was placed further downstream and the reflected waves take longer time to return. They neither appear in OpenFOAM, where the wave dissipation

of IHFOAM follows the shallow water approximation, and it is thus more effective for long waves than the beach of the physical experiment, which usually performs best for short waves.

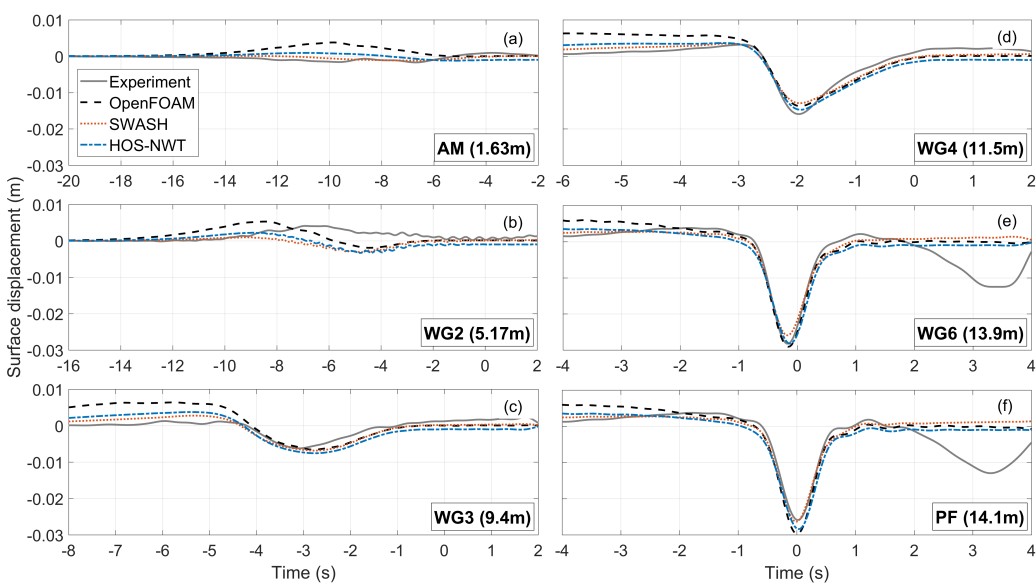

**Figure 10.** Comparison of the free surface elevation of the 2nd order difference harmonics between the numerical models and the experiment at different locations (**a**–**f**).

It should be noted that the reproduction of long waves is challenging at physical and numerical flumes, because they can be contaminated by spurious waves, which are caused by even subtle movement of the wavemaker (free displacement waves) or by the linear wave generation. Moreover, due to their large wavelength, their absorption is not trivial and may cause sloshing effects in relatively short time in a bounded domain [62]. A 2nd order wave generation can be effective in at least decreasing the spurious preceding crest of the long bound waves [19]. To further examine the 2nd order harmonics, the present results are compared with the analytical solution in the Appendix A.

*4.5. Evolution of the 3rd Order Harmonics*

The reproduction of these high frequency waves is challenging numerically and experimentally due to dissipation effects. NWTs should be well converged in order to propagate accurately short, low amplitude waves. Indeed, previous studies with "skilled" NWTs [22,26] showed considerable discrepancies for the high order harmonics, but Vyzikas et al. [5] demonstrated that a well-converged computational domain can result in accurate propagation of up to 4th order harmonics. The 4th order harmonics have similar behaviour to the 3rd order harmonics [5], and here, only the evolution of the latter is presented.

The evolution of the extracted 3rd order harmonics is shown in Figure 11. It can be seen that close to the wavemaker (Figure 11a,b), where the wave group is dispersed and not very steep, the magnitude of the 3rd order harmonics is negligible. Moreover, close to the boundary at AM, the agreement among the models and the experiment is not very good due to the spurious high order free waves. However, at downstream locations and towards the focusing of the wave group, the comparison among the models and the experiment improves considerably, with only some minor discrepancies being noticeable at the central crest and adjacent troughs.

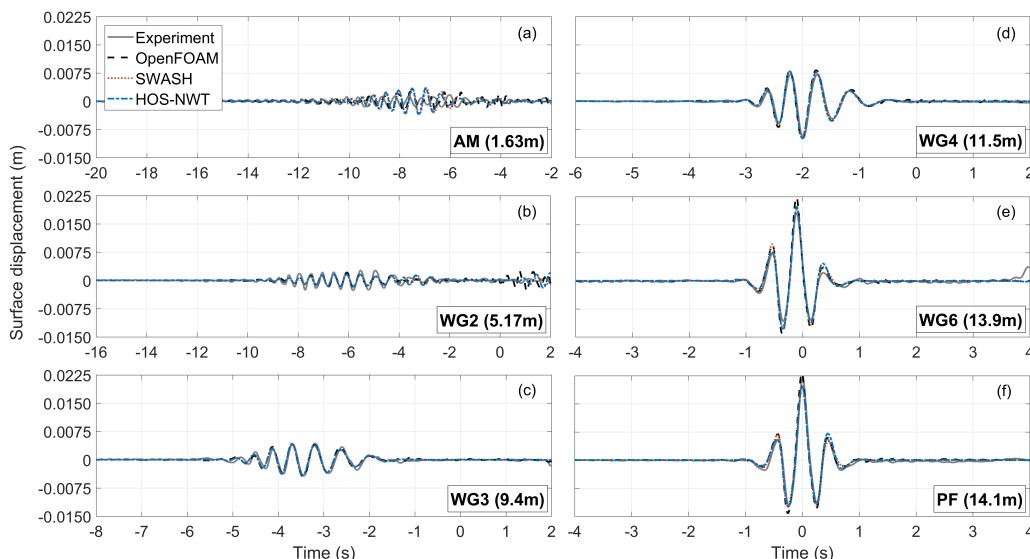

**Figure 11.** Comparison of the free surface elevation of the 3$^{rd}$ order harmonics between the numerical models and the experiment at different locations (**a**–**f**).

The accurate reproduction of high order harmonics, apart from the underlying physics of the propagation of a wave group, is also important from an engineering point of view, because they can cause dynamic excitation of marine structures. This phenomenon is referred to as "ringing" and it can cause fatigue and potential failures to offshore structures [63].

### 4.6. Wave Group Evolution

Having presented the evolution of the extracted individual harmonics up to 3$^{rd}$ order, here, the evolution of the wave group is examined from the AM to the PF location. This refers to the free surface elevation as measured in the flumes without any processing and it is presented in Figure 12. It can be seen that all the models are in good agreement with the experiment at all locations. The worst agreement is observed at WG2 in Figure 12b, where the spurious free waves have started separating from the main wave group. After the separation of the spurious waves, the comparison between the models is immediately improved. The best agreement among all the models seems to be at the middle of the tank, namely at WG3 and WG4, where influence of the boundary fades (spurious waves) and the steepness is not yet very high.

To conclude, the comparisons in this section demonstrate that, despite the considerably different versions of the governing equations and numerical approaches in OpenFOAM, SWASH and HOS-NWT, all models show exceptional performance for the propagation of very steep focused wave groups.

### 4.7. Comparison at the Focal Point

The PF location is, in most of the cases, the location of interest, where the structure is placed. For this reason, more detailed comparisons are presented here including the extracted 4$^{th}$ and 5$^{th}$ order harmonics and quantitative comparison at wave crest.

The measured surface elevation is shown in Figure 13. It can be seen that the best agreement with the experiment at the crest elevation is achieved by OpenFOAM, while SWASH and HOS-NWT underestimate the maximum elevation. The experiment shows an increased elevation after the main crest, probably due to the fact that the breaking of the wave is close to starting. The NWTs either not include breaking (HOS-NWT and SWASH) or they are not optimised for it (OpenFOAM). Despite the small differences at the wave crest, the overall shape of the wave group is very well reproduced, with the adjacent troughs and lateral crests being in excellent agreement. Noticeable differences are observed

before and after the main group, from $-2.5$ s to $-1.5$ s and from $+1.7$ s to $+2.5$ s, where the experimental surface elevation is lower than that of the numerical models.

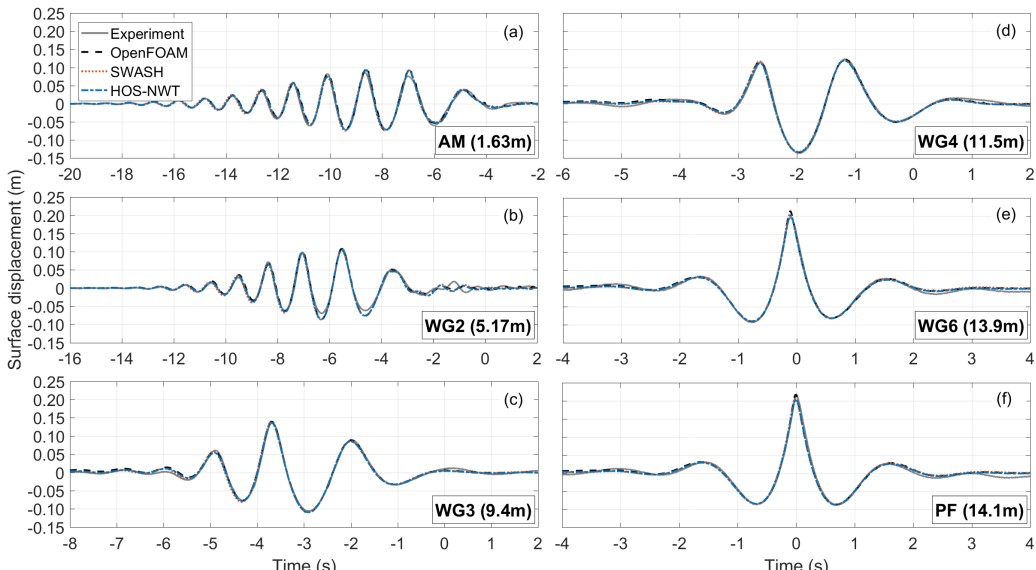

**Figure 12.** Comparison of the measured surface elevation between the numerical models and the experiment at different locations (**a**–**f**).

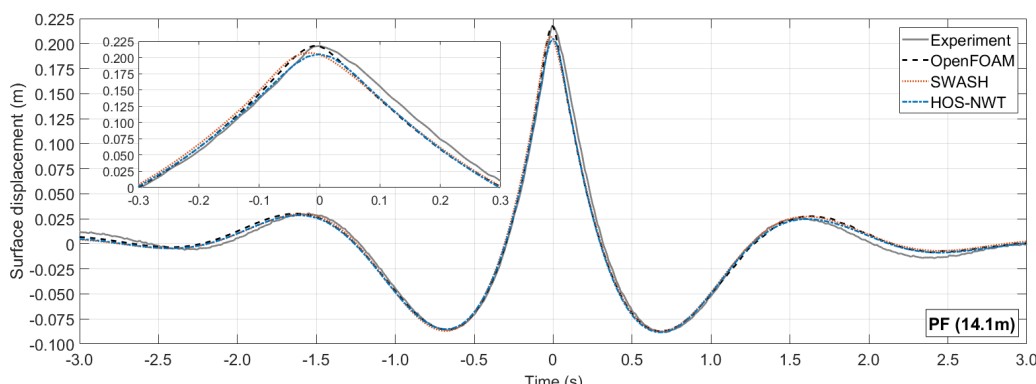

**Figure 13.** Comparison of the free surface elevation between the numerical models and the experiment at the PF location.

Next, the analysis of the individual harmonics up to 5$^{th}$ order is examined in Figure 14 at the PF location. It is noted that the 4$^{th}$ and 5$^{th}$ order harmonics are not directly obtained by the four-wave harmonic decomposition, but the former are included in 2$^{nd}$ difference harmonics and the latter in linear and 3$^{rd}$ order harmonics, as shown in Equation (4). The 4$^{th}$ order harmonics can be trivially separated with frequency filtering from the 2$^{nd}$ difference harmonics, since they occupy non-overlapping frequency bands. The 5$^{th}$ order harmonics are taken only from the linear harmonics using a similar frequency filtering at approximately $4f_p$. For simplicity, the 5$^{th}$ order harmonics within the 3$^{rd}$ order harmonics are ignored, because the magnitude of the 5$^{th}$ order harmonics within the extracted linear harmonic is much greater than that included in the 3$^{rd}$ harmonics.

Figure 14 shows that all models are in very good agreement with the experiment even for the highest order harmonics. The greatest discrepancies are observed for the 2$^{nd}$ difference harmonics and the 5$^{th}$ order harmonics. Discrepancies are observed also at the crest of the linear harmonics, with the experiment having the highest crest elevation. It is very interesting to observe that OpenFOAM appears to overestimate all the nonlinear

harmonics, but yet to give the best overall result (see Figure 13). At the same time, SWASH and HOS-NWT seem to give a very good agreement with the experiment for all the nonlinear harmonics.

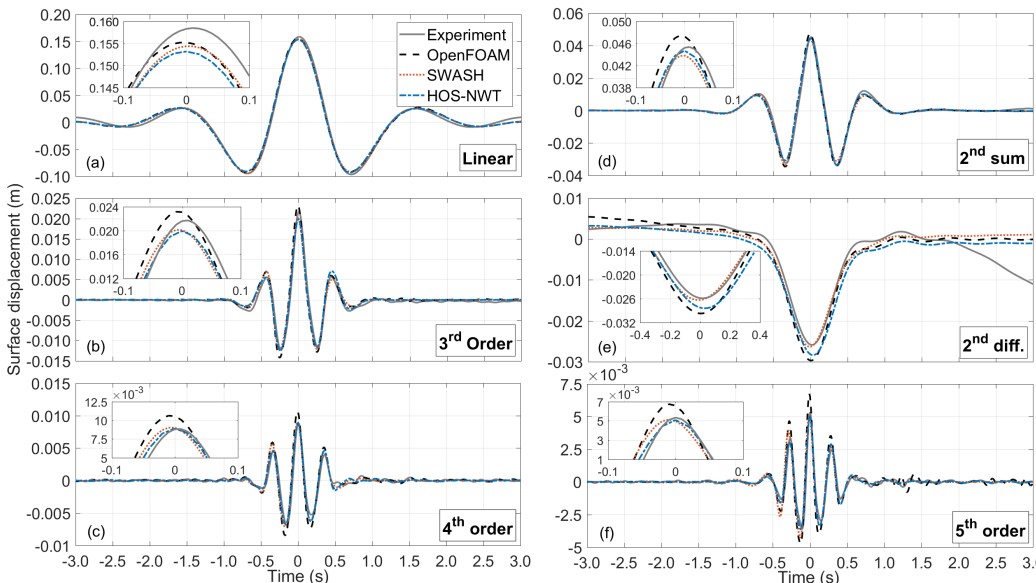

**Figure 14.** Comparison of the free surface elevation of the harmonics between the numerical models and the experiment at the PF location.

The quantitative comparison of the surface elevation at the highest crest and deepest trough (for the $2^{nd}$ difference harmonics) is presented in Table 5. It can be seen that OpenFOAM has practically an identical crest elevation to the experiment with a difference of only 0.1%, which is less that the accuracy of the experimental WGs ($\pm 1$ mm). The performance of SWASH and HOS-NWT is also impressive, since they achieve a small error of approximately 5% at the crest of nearly breaking wave groups. However, Table 5 also demonstrates that the excellent agreement between OpenFOAM and the experiment is a result of intercancellation of the overestimation of the nonlinear harmonics. In fact, for almost all of the individual harmonics, with the exception of the $3^{rd}$ order harmonics, SWASH and HOS-NWT provide the best agreement with the experiment.

**Table 5.** Intercomparison of phase-resolving models at the PF location at the crest and through ($2^{nd}$ diff). The experiment is used as the benchmark and the differences are expressed as absolute (mm) and percentage (%).

| Harmonics | Experiment | OpenFOAM | | SWASH | | HOS-NWT | |
|---|---|---|---|---|---|---|---|
| Total (measured) | 217.5 | 0.2 | 0.1% | −10.5 | −4.8% | −12.5 | −5.7% |
| Linear | 158.5 | −3.3 | −2.1% | −4.1 | −2.6% | −5.3 | −3.3% |
| $2^{nd}$ sum | 45.4 | 2.1 | 4.5% | −1.6 | −3.4% | −0.7 | −1.5% |
| $2^{nd}$ difference | −25.9 | −3.8 | 14.8% | −0.5 | 1.8% | −2.5 | 9.7% |
| $3^{rd}$ order | 21.7 | 1.5 | 6.7% | −1.6 | −7.3% | −1.8 | −8.1% |
| $4^{th}$ order | 8.8 | 1.8 | 20.2% | 0.2 | 2.1% | 0.1 | 1.6% |
| $5^{th}$ order | 5.3 | 1.4 | 26.1% | −0.2 | −3.8% | −0.3 | −4.8% |
| Sum of harmonics | 213.9 | −0.4 | 0.2% | −7.7 | −3.6% | −10.3 | −4.8% |

## 5. Conclusions

The present work examined the applicability and performance of three open-source and widely used solvers (OpenFOAM, SWASH and HOS-NWT) for the propagation of steep focused wave groups. The models lie on different formulations of the governing equations of fluid motion and employ fundamentally different methods to resolve them.

To the best of the authors' knowledge, the models have not been compared in the past under the exact same conditions, neither all of them have been validated against experiments for steep focused waves and for the reproduction of high order harmonics.

The present findings demonstrate that all the models, after thorough convergence of the NWTs and use of the focusing methodology, show very good performance in simulating the propagation of the wave group. This shows that the nonlinear near-resonant interactions and the bound wave interactions up to $5^{th}$ order are captured accurately. Especially impressive was the performance of the weakly nonlinear solvers, SWASH and HOS-NWT, clearly demonstrating that they can simulate high order nonlinear wave-wave interactions with accuracy. The best comparison with the experimental results was observed for OpenFOAM, achieving practically absolute agreement with the experiment at the crest elevation. However, it is interesting to observe that this excellent agreement stems to an extent from intercancellation of the individual nonlinear harmonics that are all overpredicted by OpenFOAM. In fact, it was shown that SWASH and HOS-NWT may provide a more accurate simulation of the individual harmonics, which, for certain types of studies, e.g., overtopping [64,65], may be important. Thus, the present results show that, for wave propagation, CFD two-phase models is not necessarily the gold standard in numerical modelling. This is a very important aspect to take into account when using CFD models to validate other numerical models. A possible explanation for this observation is that SWASH and HOS-NWT are numerical models specifically developed to simulate waves, while OpenFOAM is a more general modelling CFD platform that was only adapted to simulate waves using appropriate boundary conditions. Also, the complication of solving for two-phase fluid flows in OpenFOAM may be a source of error that should not be factored out. To confirm these findings and be able to identify credibly the sources of the errors, other suitable solvers can be used that employ similar equations and modelling approaches as these of the present work [66,67].

The next future step in order to establish a robust validation for studying extreme waves numerically using the present methods is to compare the three models with results for the kinematics under steep focused wave groups from analytical solutions [68] or experiments [69,70]. This will set the basis for further studying the loading on structures with a design wave that will be representative of a steep extreme wave event. Already however, the present findings open the possibility for coupling of the models, for example combination of PFT for the far field propagation and CFD near the examined structure [16,71], or for using SWASH and HOS-NWT for preliminary investigations and a computationally efficient correction of the phases and amplitudes of the wave group using the focusing methodology.

**Author Contributions:** Conceptualization & methodology: T.V. and D.S.; software, validation, formal analysis, writing—original draft preparation, writing—review and editing: T.V.; supervision: D.G. and C.M. All authors have read and agreed to the published version of the manuscript.

**Funding:** The authors would like to thank the School of Engineering of the University of Plymouth, Ifremer, LabEx Mer and the EPSRC SuperGen project SMARTY (grant number EP/J010316/1) for the financial support.

**Conflicts of Interest:** The authors declare no conflict of interest. The funders had no role in the design of the study; in the collection, analyses, or interpretation of data; in the writing of the manuscript, or in the decision to publish the results.

**Abbreviations**

The following abbreviations are used in this manuscript:

AM  Amplitude matching location
CF  Crest-focused wave group
CFD  Computational Fluid Dynamics
FFT  Fast Fourier Transform
FVM  Finite Volume Method
HOS  High Order Spectral method
MWL  Mean water level
NSE  Navier-Stokes Equations
NSWE  Nonlinear Shallow Water Equations
NWT  Numerical Wave Tank
PF  Phase focal location
PFT  Potential Flow Theory
PM  Pierson-Moskowitz Spectrum
RANS  Reynolds Averaged Navier-Stokes equations
SWL  Still water level
TF  Trough-focused wave group
VoF  Volume of Fluid method
WG  Wave gauges

## Appendix A. Comparison with 2nd Order Theory

In this section, the 2nd order harmonics are further analysed and compared with the analytical "exact" solution. In particular, based on the analysis in Section 4.4, the 2nd order difference harmonics exhibited the greatest discrepancies between the numerical models and the experiments, due to spurious effects from the wave generation. The comparison with the analytical solution can indicate which wave generation method and model is more accurate for the 2nd order harmonics.

The analytical expression for the 2nd order solution of an irregular wave signal was given for two waves by Dalzell [72]. Here, the expressions presented refer to an arbitrary number of linear wave components ($N$), propagating in a single direction on finite water depth. Dalzell's approach follows the same potential flow theory assumptions as [73], but it is based on symbolic computations. Its application is straightforward and it was used for the 2nd order wave generation boundary conditions in OpenFOAM [74].

According to 2nd order theory, the surface elevation $\eta$ is given from the first harmonic and the matrix of the 2nd order interactions with each free wave with all the other free waves, including the self-interaction, as shown in Equation (A1). The 2nd order harmonics are calculated by Equations (A2) and (A3). The respective coefficients are given in Equations (A4) and (A5) for any possible combination of any wave component $i$ with a component $j$. Since, a unidirectional wave propagation is assumed for the present case, the angle between the components is zero and the $\cos(\phi_i - \phi_j) = 1$.

$$\eta = \sum_{i=1}^{N} \alpha_i \cos \psi_i + 2^{\text{nd}} \text{ sum } + 2^{\text{nd}} \text{ difference} \tag{A1}$$

where the phase function of a wave $i$, $\psi_i = k_i x - \omega_i t + \epsilon_i$, with $\epsilon_i$ the arbitrary phase of a wave, $\alpha_i$ is the wave amplitude of first order.

$$\begin{aligned} 2^{\text{nd}} \text{ sum } = {} & \sum_{i,j=1}^{N} \frac{\alpha_i^2 |k_i|}{4 \tanh(|k_i|h)} \left[ 2 + \frac{3}{\sinh^2(|k_i|h)} \right] \cos(2\psi_i) \\ & + \alpha_i \alpha_j B_p(k_i, k_j) \cos(\psi_i + \psi_j) \end{aligned} \tag{A2}$$

$$2^{\text{nd}} \text{ difference } = \alpha_i \alpha_j B_m(k_i, k_j) \cos(\phi_i - \psi_j) - \sum_{i,j=1}^{n} \frac{a_i^2 |k_i|}{2 \sinh(2|k_i|h)} \tag{A3}$$

The solution above is given for non-zero MWL, which corresponds to the last term in Equation (A3). In the present case, the deviation of the MWL from zero is negligible, since the volume of water in the physical wave flume is constant and the numerical simulations, especially the CFD two-phase simulation in OpenFOAM, do not have a long duration that may result in higher volume of water.

$$
\begin{aligned}
B_p(k_i, k_j) = & \frac{\omega_i^2 + \omega_j^2}{2g} - \frac{\omega_i \omega_j}{2g} \left[ 1 - \frac{1}{\tanh(|k_i|h) \tanh(|k_j|h)} \right] \\
& \times \left[ \frac{(\omega_i + \omega_j)^2 + g|k_i + k_j| \tanh(|k_i + k_j|h)}{D_p(k_i, k_j)} \right] \\
& + \frac{\omega_i + \omega_j}{2g D_p(k_i, k_j)} \left[ \frac{\omega_i^3}{\sinh^2(|k_i|h)} + \frac{\omega_j^3}{\sinh^2(|k_j|h)} \right]
\end{aligned} \tag{A4}
$$

$$
\begin{aligned}
B_m(k_i, k_j) = & \frac{\omega_i^2 + \omega_j^2}{2g} + \frac{\omega_i \omega_j}{2g} \left[ 1 + \frac{1}{\tanh(|k_i|h) \tanh(|k_j|h)} \right] \\
& \times \left[ \frac{(\omega_i - \omega_j)^2 + g|k_i - k_j| \tanh(|k_i - k_j|h)}{D_m(k_i, k_j)} \right] \\
& + \frac{\omega_i - \omega_j}{2g D_m(k_i, k_j)} \left[ \frac{\omega_i^3}{\sinh^2(|k_i|h)} - \frac{\omega_j^3}{\sinh^2(|k_j|h)} \right]
\end{aligned} \tag{A5}
$$

where the functions $D_p(k_i, k_j)$ and $D_m(k_i, k_j)$ are defined as:

$$D_p(k_i, k_j) = (\omega_i + \omega_j)^2 - g|k_i + k_j| \tanh(|k_i + k_j|h) \tag{A6}$$

$$D_m(k_i, k_j) = (\omega_i - \omega_j)^2 - g|k_i - k_j| \tanh(|k_i - k_j|h) \tag{A7}$$

To find the $2^{\text{nd}}$ order harmonics analytically using the previous formulas, the linear harmonics should be known. In the present study, this can be extracted accurately at any location in the flume, thanks to the four-wave decomposition. As shown in [5,75], which followed a similar principle to that of [7], for the better estimation of the $2^{\text{nd}}$ order harmonics, the evolved (locally broadened) linear harmonic should be considered. Using the harmonic from the original spectrum (see Table 2) would result in considerably higher discrepancies in the comparisons since the underlying spectra are different. Here, the comparisons will be performed for the wave group at PF, where the nonlinear harmonics reach their maximum energy content, and the spurious bound waves have separated from the main wave group, as well as any reflections have not returned in the examined time window. For simplicity, and since the extracted linear harmonics are almost identical among the numerical models and the experiment, the extracted harmonic from HOS-NWT is used, which also had the best agreement with the target spectrum at AM, as shown in Figure 6a.

The analytically calculated $2^{\text{nd}}$ order harmonics are compared with the extracted $2^{\text{nd}}$ order harmonics at PF from the nonlinear models and the experiment in Figure A1. The quantitative comparison of the crests and adjacent troughs of the $2^{\text{nd}}$ sum harmonics, as well as the trough of the $2^{\text{nd}}$ difference harmonics, is presented as (%) difference between the corresponding values of the analytical solution and the extracted harmonics ($\frac{|\text{measured - theory}|}{\text{theory}}$) in Table A1. It is noted that the adjacent troughs of the $2^{\text{nd}}$ sum harmonics are not identical, and for the comparisons of Table A1, their mean value is considered.

For the $2^{\text{nd}}$ sum harmonics, Figure A1a shows that the extracted harmonics of the experiment and the models have a crest which is similar, but higher than that of the

analytical 2nd order solution. Moreover, the extracted harmonics of the models and the experiment produce deeper lateral troughs compared to analytical solution.

For the 2nd difference harmonics, Figure A1b the analytical solution predicts a shallower trough than that of the extracted harmonics. The extracted harmonics that are the closest to the theoretical results are reproduced by the experiment and SWASH. Open-FOAM predicts the deepest trough, followed by HOS-NWT. It can be also observed that, in contrast to the experimental and the numerical harmonics, the analytical solution of the 2nd order difference harmonics do not include the spurious preceding set-up, which can be spotted between the times $-2$ s to $-1$ s in Figure A1b.

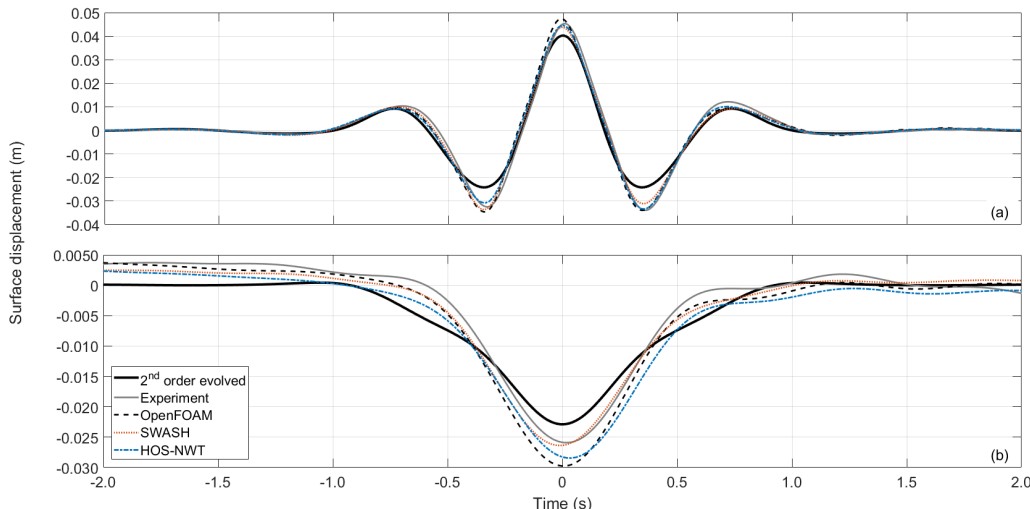

**Figure A1.** Comparison of the free surface elevation of the calculated and extracted 2nd order harmonics at the PF location: (**a**) 2nd sum; (**b**) 2nd diff.

The quantitative comparison of Table A1 demonstrates that the crest of the 2nd order sum harmonics and the trough of the 2nd order difference harmonics can be reasonably predicted by 2nd order theory, when the evolved extracted linear amplitude spectrum is used. It can be seen that OpenFOAM gives the greatest overprediction of the 2nd order harmonics compared to the analytical solution, while SWASH is the model that gives the best agreement with the analytical 2nd order solution. The better performance of SWASH and HOS-NWT compared to OpenFOAM can be justified from the fact that the two former models are specifically designed for the propagation of non-breaking waves, as those examined in the present study, while the VoF can introduce complications and numerical artefacts [62].

**Table A1.** Comparison between the extracted 2nd sum and diff harmonics with 2nd order theory solution using the evolved extracted linear amplitude spectrum at PF.

|  | Sum Crest | Sum Trough | Diff Trough |
|---|---|---|---|
| 2nd order theory calculated (mm) | +40.3 | −24.2 | −22.9 |
| Experiment | +12.7% | −37.3% | −13.2% |
| OpenFOAM | +17.8% | −41.5% | −30.0% |
| SWASH | +8.8% | −33.7% | −15.3% |
| HOS-NWT | +11.0% | −33.8% | −24.3% |

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
