# Peer review of "Intercomparison of Three Open-Source Numerical Flumes for the Surface Dynamics of Steep Focused Wave Groups"

_fluids, doi:10.3390/fluids6010009_

Round 1

Reviewer 1 Report

This manuscript presents an evaluation of three wave-resolving models with different set of equations applied to the propagation and focus of steep wave groups, involved in freak wave generation. The models are: a 2D-vertical two-phase Navier-Stokes Equations (NSE) model (OpenFOAM), a 2DV free-surface NSE model (SWASH) and 1DH fully nonlinear potential-flow model (HOS-NWT). The three models are compared with an experiment in a 20-m long wave tank, where a paddle produces a wave group that focuses somewhere in the tank. The evaluation is based on harmonics extraction and on a focusing methodology where harmonic phases and amplitudes of each model are corrected for optimal focus at the desired location. For fair comparison, model convergence with respect to horizontal and vertical resolution is wanted and an optimal configuration is selected in each case. It is found that all models perform very well but the two-phase model, even though it is considerably more expensive to run than the two others (by 4 to 5 orders!), has the least accurate solution for nonlinear harmonics (with compensating errors in the total signal). This is an important result, considering the current competition between two-phase flow models (that can explicitly solve overturning breaking waves) and single-valued free-surface models that treat the surface boundary of non-breaking waves more accurately (but consider the breaker front as a dissipating bore).

The manuscript is well written and referenced; the methods used to setup the comparison and perform the analyses are detailed and rigorous (based on previous publications by the authors).  Here, the experimental setup has been improved compared to previous work and the model intercomparison is presented as a main target (including computational cost, which is an important aspect). I think that the manuscript will be useful to the wave modeling community and can be published with minor corrections, after considering my concerns below (and few specific remarks that follows):

  1. Comparison methodology. If iterative corrections are made to allow the wave group in all models to focus at the measured focal point (instead of translating the focal point after the simulations during the comparison process), can we expect this correction to be effective not only on errors produced by the numerical wavemakers (as intended), but also on propagation errors? If so, it would not be surprising that all the models perform well and that their residual errors are marginal. The intercomparison would remain valid but possibly underestimate model errors. We may even wonder if the cancellation of errors in OpenFOAM is a result of the focusing methodology. Some comment would be useful.
  2. SWASH description. There is confusion about the description of SWASH here. It is first described (eq. 8 and 9) as a 1DH nonlinear and dispersive model described in Zijlema et al. (2011), while we learn soon after that the model actually used is the multi-layered (2DV) version described in Zijlema and Stelling (2005, 2008). The correct equations and description should be presented. It would actually be interesting to compare the two versions as the multi-layered model is expected to improve the model’s frequency dispersion.
  3. The discussion section could make better use of the numerical characteristics of the different models. SWASH is a low-order accurate finite-difference model with semi-implicit time integration, which is expected to add considerable diffusion. Is dissipation important in the simulation and does it contribute to more realism or less; in other words: are non-breaking waves really inviscid? Another question is: why does OpenFOAM show larger errors in nonlinear harmonics? Is it related to the inaccurate location of the free surface in a Cartesian two-phase model (and associated inaccuracy of surface boundary conditions) or something else?

Specific remarks:

  • Eq. 5, page 8: phase was psi in Eq. 1 then becomes phi in Eq. 4 and 5?
  • Eddy viscosity, page 10: What would be the role of non-breaking wave viscosity? And that of numerical diffusion/dispersion errors?
  • Open boundary conditions, page 11: “weakly reflective boundary condition” refers to long waves in this model I think.
  • Table 4, page 14: why is swash presented as 2D and OpenFOAM as 3D? See my main remarks.
  • L 502, page 14: observed => observe

Author Response

Thank you for your valuable comments.

Reviewer 2 Report

This paper presents an extensive analysis of a problem well known in the CFD field. Results confirm the validity of the coupled model approach although there is still some room for improvement in the multiphase solvers field. IT is worth citing "Nelli, F., Bennetts, L., Skene, D., & Toffoli, A. (2020). Water wave transmission and energy dissipation by a floating plate in the presence of overwash" where acoupled HOS-NWT model was used to simulate wave-floating body interaction.

Author Response

Thank you for your comments.

Reviewer 3 Report

The authors presented a comparative numerical investigation of focusing extreme waves and intercompared different numerical models against the experiment, especially regarding the evolution of different harmonics. The chosen numerical models are representative, the experimental setup and process is well explained and the comparison and discussion is in depth and convincing. The reviewer regards the manuscript eligible for publication in Fluids, given the authors address the following comments:

  • In section 3.2, the authors described the set-up of the SWASH NWT. 6 vertical layers have been used for the simulation while a further test of 8 vertical layers is used. The authors mentioned that this set-up is of much higher resolution than the typical 1-2 layers set-up. The question is: why the authors choose not to use the 2-layer set-up, what is the challenge, why starting from 6 layer and even use finer layers, what is the motivation. The convergence only compared 6 and 8 layers, which is not sufficiently convincing. As far the reviewer is aware, each increase of vertical layers leads to an increase of magnitude of computational time. The kpd value is only 1.75, it is not an extremely deep-water condition. Thus, the use of extensive vertical layers especially needs further explanation.

  • The authors have chosen the numerical models wisely in general and did good comparison. However, the choice of the potential flow model might not be as representative. Finite difference based sigma-grid potential flow models such as OceanWave3D [1, 2] and REEF3D::FNPF [3] are also fast potential flow models and can produce much steeper waves and even approximate strong breaking waves. They can be better candidates for the steep focus wave tests, such as being done in [4]. Further tests using these models are not required by the authors since the learning process of the models takes time, but it is highly recommended that the authors refer to these models in the introduction section or future work section to give a more state-of-art overview and a complete scope of the relevant research.

  • The focal point surface elevations are in general well achieved by all tested models. However, since the authors mention that it is a near breaking situation at the focal point, then it is important to give a bit more information about the breaking algorithms and turbulence models in the numerical models. It is recommended to include information on: what turbulence model is used in OpenFOAM? whether the parametrized breaking algorithm in SWASH (and maybe HOS-NWT) is turned on, and what are the parameters (breaking wave steepness criterion and strength of energy dissipation). No details on the turbulence model and breaking algorithm are required, but they need to be explicitly shown to the readers.

References:

[1] Engsig-Karup, A., Bingham, H. and Lindberg, O. (2009). An efficient flexible-order model for 3D nonlinear water waves. Journal of Computational Physics, 228, 2100–2118.

[2] Engsig-Karup, A.P., Madsen, M.G. and Glimberg, S.L. (2012). A massively parallel GPU-accelerated model for analysis of fully nonlinear free surface waves. International Journal for Numerical Methods in Fluids, 70(1).

[3] Bihs H., Wang W., Martin T. and Kamath A. (2020) REEF3D::FNPF - A Flexible Fully Nonlinear Potential Flow Solver. Journal of Offshore Mechanics and Arctic Engineering. 142(4).

[4] Wang W., Kamath A., Pakozdi C. and Bihs H. (2019) Investigation of Focusing Wave Properties in a Numerical Wave Tank with a Fully Nonlinear Potential Flow Model. Journal of Marine Science and Engineering, 2019, 7, 375.

Author Response

Thank you for your comments.
